# Effects of Chromium Propionate and Calcium Propionate on Lactation Performance and Rumen Microbiota in Postpartum Heat-Stressed Holstein Dairy Cows

**DOI:** 10.3390/microorganisms11071625

**Published:** 2023-06-21

**Authors:** Chenxu Zhao, Bingyu Shen, Yan Huang, Yezi Kong, Panpan Tan, Yi Zhou, Jiaqi Yang, Chuang Xu, Jianguo Wang

**Affiliations:** 1College of Veterinary Medicine, Northwest A&F University, Yangling 712100, China; cxzhao@nwafu.edu.cn (C.Z.); shenby@nwafu.edu.cn (B.S.); hy040016@nwafu.edu.cn (Y.H.); kongyezi0207@163.com (Y.K.); tanpp@nwsuaf.edu.cn (P.T.); zhouy1@nwafu.edu.cn (Y.Z.); yjiaqi@nwafu.edu.cn (J.Y.); 2College of Animal Science and Veterinary Medicine, Heilongjiang Bayi Agricultural University, Daqing 163000, China; 3College of Veterinary Medicine, China Agricultural University, Beijing 100193, China

**Keywords:** chromium propionate, calcium propionate, heat stress, rumen microbiota, dairy cows

## Abstract

Chromium propionate (Cr-Pro) and calcium propionate (Ca-Pro) are widely applied in dairy production, especially in the alleviation of heat stress (HS). HS can reduce the abundance of rumen microbiota and the lactation performance of dairy cows. The present work mainly focused on evaluating the effects of Cr-Pro and Ca-Pro on the performance, ruminal bacterial community, and stress of postpartum HS dairy cows as well as identifying the differences in their mechanisms. Fifteen multiparous postpartum Holstein cows with equivalent weights (694 ± 28 kg) and milk yields (41.2 ± 1.21 kg/day) were randomly divided into three groups: control (CON), Cr-Pro (CRPR), and Ca-Pro (CAPR). The control cows received the basal total mixed ration (TMR) diet, while the CRPR group received TMR with 3.13 g/day of Cr-Pro, and the CAPR group received TMR with 200 g/day of Ca-Pro. The rumen microbial 16S rRNA was sequenced using the Illumina NovaSeq platform along with the measurement of ruminal volatile fatty acids (VFAs) and milking performance. Cr-Pro and Ca-Pro improved lactation performance, increased the rumen VFA concentration, and altered the rumen microbiota of the HS dairy cows. Cr-Pro significantly improved the milk yield (*p* < 0.01). The richness and diversity of the microbial species significantly increased after feeding on Ca-Pro (*p* < 0.05). Gene function prediction revealed increased metabolic pathways and biological-synthesis-related function in the groups supplemented with Cr-Pro and Ca-Pro. Our results indicate that the application of Cr-Pro or Ca-Pro can provide relief for heat stress in dairy cows through different mechanisms, and a combination of both is recommended for optimal results in production.

## 1. Introduction

With the increasing high temperatures and duration of summer, the negative effects of heat stress (HS) on dairy cows are becoming more serious. Compared to other farm animals, cows are more sensitive to HS [1,2]. At present, a temperature–humidity index (THI) of 72 is used to determine HS, but for high-yielding lactating cows, it is 68. HS can reduce feed intake, digestive efficiency, milk production, and milk quality in dairy cows. It also results in negative energy balance and impairs reproductive performance, leading to significant economic losses [1,3]. Despite the widespread use of multiple physical cooling measures such as shade, fans, and spraying, HS still incurs an estimated annual cost of over USD 1.2 billion in the United States [4]. In addition, HS also affects the structure and metabolism of the ruminal microbiota, causing lactic acid accumulation in the rumen [5], a decrease in pH, and an increasing risk of rumen acidosis [2]. Given the detrimental impact of HS on the sustainable development of the dairy industry, new effective prevention and control strategies are urgently needed.

Propionic acid, one of the components of VFAs, is produced through rumen microbial fermentation and serves as a precursor for glucose production, providing up to 90% of the body’s glucose [6,7]. In recent years, propionate has been increasingly used in dairy farming. Notably, calcium propionate (Ca-Pro) is widely used as a preservative in food, feed, and medicine due to its efficacy against mycetes and bacteria and its neutral taste and flavor [8]. Additionally, Ca-Pro can alleviate the negative energy balance and hypocalcemia in postpartum cows [8]. In the rumen, Ca-Pro is hydrolyzed to propionate and Ca^2+^, and both are important components of rumen fluid. Studies have confirmed that Ca-Pro can improve ruminal energy metabolism and promote rumen development [6,9]. Furthermore, chromium propionate (Cr-Pro), another form of propionate, has been gradually used as a feed additive. It plays a crucial role in the metabolism of carbohydrates, lipids, and proteins [10,11] and regulates energy metabolism by influencing insulin levels [12]. Studies have shown that chromium supplementation improves milk yield in early lactation (21 days after delivery) and modulates immunity and reproduction in dairy cows [13,14,15,16]. Pechova (2018) proposed that chromium can alleviate various stresses induced in animals, encompassing physiology, pathology, nutrition, and the environment [11]. Al-saiady et al., (2004) observed that chromium can improve milk yield of HS cows without affecting milk quality [17]. The use of chromium is limited due to toxicity. Compared to non-toxic trivalent chromium, which is considered safe, hexavalent chromium, discovered earlier and studied extensively, is known to cause multiorgan cancers [10,11]. Therefore, Cr-Pro is the only source of chromium certified by the U.S. Food and Drug Administration (FDA) that can be added to cattle and sheep feed, with a limited supplemental dose of 0.5 mg/kg [14]. Compared to Cr-Pro, Ca-Pro is more widely used because it is odorless and non-toxic, particularly as a food and feed additive [7]. An increasing number of investigations have demonstrated the beneficial effects of Ca-Pro and Cr-Pro on energy metabolism, growth, and reproductive performance [6,7,8,9,10,11,12,13,14,15,16]. However, their effects on the production, immune regulation, and rumen microbiota of postpartum HS dairy cows have rarely been reported. This study aimed to investigate whether Cr-Pro and Ca-Pro can regulate the stress, immunity, and performance of postpartum HS cows by modulating the rumen microbiota, providing further scientific basis for their application in production.

## 2. Materials and Methods

### 2.1. Ethics Statement

The experimental designs and protocols adopted in this work were approved by the Animal Ethical and Welfare Committee, Northwest A&F University (approval no. 2020078), and were conducted in accordance with the university’s guidelines for animal research (protocol number NWAFAC1008).

### 2.2. Animals, Diets, and Experimental Design

In total, 15 multiparous (2–4 parity) lactating Holstein dairy cows (32 ± 4 days after delivery) with no clinical signs of disease were sourced from a commercial 7000 head dairy farm located in Nenjiang (Heilongjiang, China). The cows were fed the same ration three times a day at 7:00 a.m., 2:00 p.m., and 6:30 p.m. and milked three times daily at 6:30 a.m., 1:30 p.m., and 9:30 p.m. The cows were provided access to water *ad libitum* during the time of the experiment (31 days in August 2020). The cows were then randomly allocated into three treatment groups: Control group (CON, *n* = 5; milk yield, 41.2 ± 0.85 kg/day), fed the basal total mixed ration (TMR) diet (Table 1); Ca-Pro supplementation group (CAPR, *n* = 5; 40.7 ± 1.38 kg/day), fed the basal TMR mixed with 200 g/day of Ca-Pro (content, 99.21%; Jiang Xinyu Biological Technology Co., Ltd., Wuhan, China); Cr-Pro supplementation group (CRPR, *n* = 5, 41.8 ± 1.21 kg/day), fed the basal TMR mixed with 10 g/day of Cr-Pro (content, 31.3%; Jiangxinyu), i.e., equivalent to 3.13 g/day of pure Cr-Pro. The fans installed in the cowshed were turned on when the temperature was higher than 21 °C.

The rectal temperature and respiratory rate were measured (8:00 a.m., 2:00 p.m., and 8:00 p.m.) at three-day intervals during the experiment, and the average of the three tests on that day was used. The automatic temperature and humidity recorder (EN100, EXTECH, Shourong Industrial Equipment Co., Ltd., Shanghai, China) was placed 1.5 m above the ground to monitor the temperature (T) and relative humidity (RH) at 8:00 a.m., 2:00 p.m., and 8:00 p.m. every day in the cowshed. The formula for calculating the temperature and humidity index was THI = 0.81 T + (0.99 T − 14.3)RH + 46.3 [3,18].

### 2.3. Sample Collection and Analysis

At the end of the experiment (day 31), ruminal contents were collected from all cows at 15 min before feeding and 4 h after feeding by a rumen catheter on the last day of the experiment. The collected ruminal contents were blended and filtered through four layers of cheesecloth. Subsequently, 5 mL samples were acidified with 100 µL of orthophosphoric acid for VFA analysis, while 10 mL samples were immediately frozen in liquid nitrogen and stored at −80 °C until DNA extraction.

Blood samples were obtained from the tail using venipuncture 30 min after feeding. The blood samples were collected in heparinized 10 mL evacuated tubes and then centrifuged at 2000× *g* for 10 min. The resulting supernatant plasma was packaged and stored at −80 ℃ for measuring the concentrations of malonaldehyde (MDA) (kit no. A003-1), total antioxidant capacity (T-AOC) (kit no. A015-2), superoxide dismutase (SOD) (kit no. A001-1), and myeloperoxidase (MPO) (kit no. A044-1) (Nanjing Jiancheng Bioengineering Institute, China). The concentrations of plasma tumor necrosis factor-α (TNF-α) (kit no. ML077389), interleukin-1β (IL-1β) (kit no. ML064295), interleukin-6 (IL-6) (kit no. ML064296), and interleukin-10 (IL-10) (kit no. ML777257) were analyzed using ELISA kits (Enzyme-linked Biotechnology Co., Ltd., Shanghai, China).

Milk samples were collected from all 15 cows three times on the 31st day. For each collection, 50 mL of milk was taken after thorough mixing. Potassium dichromate (30 mg) was added to the milk samples, mixed evenly, and stored at 4 ℃. These samples were used to determine the protein, fat, and urea nitrogen (MUN) contents of the milk using a milk composition analyzer (LACTOSCAN S60-M, MILKOTRONIC Co., Ltd., Nova Zagora, Bulgaria). Milk production was recorded using the Afimilk MPC system (Afimilk Agricultural Science and Technology Co., Ltd., Kibbutz Afikim, Israel). The recorded value represents the average of three measurements taken on the same day. The average milk yield of the first three days of the experimental period was calculated as the initial milk yield of the cows, while the average milk yield of the last three days was used for comparison of the results.

### 2.4. Volatile Fatty Acids Measurement

The ruminal contents were centrifuged for 10 min at 4000 rpm. Then, 1 mL of the supernatant was transferred into a 2 mL centrifuge tube. To this, 0.6 mL of 50% sulfuric acid solution containing 50 μL of the internal standard cyclohexanone was added. Subsequently, 4 mL of ether was added for shock extraction at 4 °C for 5 min, followed by centrifugation at 10,000× *g* for 5 min. The resulting supernatant was measured using a gas chromatograph (QP2010 Ultra, Shimadzu, Japan), equipped with a WAX capillary column (30 m × 0.25 mm × 0.25 μm). The column was initially maintained at a constant temperature of 40 °C for 3 min and then increased to 210 °C for 5 min at a rate of 30 °C/min. The injector temperature was set at 220 °C, and the detector temperature at 230 ℃. The helium carrier gas flow rate was maintained at 1.0 mL/min. Data were collected in single ion monitoring (SIM) mode with a sampling interval of 0.3 s.

### 2.5. DNA Extraction and Sequencing

Approximately 10 mL of the rumen content from each sample were stored in dry ice and sent to Shanghai Personal Biotechnology Co., Ltd. (Shanghai, China) for detection of the ruminal microbes’ 16S rRNA. Total microbial DNA was extracted from each rumen sample using the QIAamp DNA Fecal Mini Kit (No. 51504, QIAGEN Co., Ltd., Shanghai, China) following the product manual. The DNA quality and quantity were measured using the NanoDrop OneC Microvolume UV-Vis Spectrophotometer (Thermo Scientific, Waltham, MA, USA) and 1.2% agarose gel electrophoresis. We amplified the 16S rRNA V3–V4 region from the bacteria using the primers 338F (5’-ACTCCTACGGGAGGCAGCAG-3’) and 806R (5’-GGACTACHVGGGTWTCTAAT-3’) with an ABI GeneAmp 2720 PCR thermocycler (ABI, Foster City, CA, USA). The PCR amplification protocol included an initial denaturation at 98 °C for 2 min, followed by 30 cycles of denaturing at 98 °C for 15 s, annealing at 55 °C for 30 s, and extension at 72 °C for 30 s; a single extension at 72 °C for 5 min, and finally at 4 °C. The reaction mixture consisted of 0.25 µL of Q5 High-Fidelity DNA Polymerase (M0491, New England Biolabs Ltd., Beijing, China) in 25 µL of 5× reaction buffer (5 μL), 5× GC buffer (5 μL), 2.5 mM dNTP (2 μL), 10 uM forward primer (1 μL), 10 uM reverse primer (1 μL), DNA template (2 μL), and ddH2O (8.75 μL). The Illumina NovaSeq platform was used for paired-end sequencing.

The sequences were filtered, denoised, merged, and checked for nonchimeric reads using the DADA2 method of QIIME2 software (v2019.4). High-quality sequences were then clustered into amplicon sequence variants (ASVs) with 100% similarity [19]. The ASVs were classified using the Ribosomal Database Project (RDP) Classifier trained on Greengenes (Release 13.8, http://greengenes.secondgenome.com/, accessed on 16 May 2021) reference database [20]. Data from the sequencing were deposited in the Sequence Read Archive (SRA) of NCBI (no. SRR416794: PRJNA922454).

### 2.6. Statistical Data Analysis

The plasma parameters (oxidation, antioxidation, and cytokines), lactation performance (milk yield, fat, protein, and MUN), and rumen fermentation parameters (VFAs) were analyzed using one-way ANOVA. Comparative analysis was conducted using the LSD postmortem test in SPSS software (version 22.0, SPSS Inc., Chicago, IL, USA). The relative abundances of bacterial taxa among the experimental groups were compared using the Kruskal–Wallis H test and the Wilcoxon rank-sum test. Alpha-diversity measurements, including richness (ASVs, phyla, and genera), Observed_species, and Shannon index, were assessed using QIIME2 (2019.4). The overall microbiotas shaped by the three treatments were compared using principal coordinates analysis (PCoA) based on Bray–Curtis distances. Permutational multivariate analysis of variance analysis (PERMANOVA), based on the Bray–Curtis distance matrices, was used to evaluate whether the rumen bacterial structures were significantly distinct between groups using the “Adonis” procedure in the R “vegan” package. Linear discriminant analysis (LDA) effect size (LEfSe) was used to compare the relative abundances of microbial taxa, with significant differences considered when the LDA score was >2 and *p* < 0.05. Functional gene prediction analysis of the 16S genes was conducted through the Kyoto Encyclopedia of Genes and Genomes (KEGG) using PICRUSt2 to predict gene functions and calculate gene abundance. Visualization was performed using STAMP v2.1.3. Spearman correlation coefficients between the rumen microbes, TVFAs, oxidative stress index, cytokines, and lactation performance were calculated using the genescloud tools (https://www.genescloud.cn, accessed on 13 September 2022).

## 3. Results

### 3.1. Heat Stress Determination

During most of the experiment, the THI of the barn ranged from 68 to 80, with an average of 73.2; therefore, the test cows were under mild HS (Figure 1A). During the later stage of the experiment, with a decrease in the THI, the respiratory rate of the cows in all groups decreased, with the CAPR group being significantly lower than the CON group (*p* = 0.012) (Figure 1C). There was no significant difference in the rectal temperature among all groups (*p* > 0.05) (Figure 1B).

### 3.2. Lactation Performance and Ruminal TVFAs

Among the three groups, the milk yield of the CRPR group was the highest (*p* < 0.01); compared to the CON group, the mean value increased by 13.3 kg/day. This was followed by the CAPR group, in which the milk yield improved by 2.5 kg/day (*p* > 0.05). Compared to the CON group, the milk fat, milk protein, and milk MUN were higher in the CAPR and CRPR groups, but the difference was not significant (*p* > 0.05). The ratio of milk fat to milk protein was 1.14 in the CAPR group, which was optimal among the three groups (Table 2).

The concentrations of acetate, propionate, butyrate, and total VFAs (TVFAs) in the CRPR group were highest among the three groups (*p* < 0.01). Meanwhile, the concentration of propionate in the CAPR and CRPR groups was higher than in the CON group (*p* = 0.064 and *p* = 0.032, respectively), and the acetate/propionate ratio was lower (Table 2).

### 3.3. Composition and Diversity of the Rumen Microbiota

A total of 1,135,009 raw reads were obtained across the three groups, and 1,019,902 valid tags were generated after screening, accounting for 89.9% of the raw reads. The average sequence length was 421 bp (Supplementary Data S1). A total of 37,971 ASVs were detected in the three groups across all samples using QIIME2 clustering, including 11,898 in the CON group, 22,075 in the CAPR group, and 14,846 in the CRPR group (Figure 2A). The number of unique ASVs in the CAPR and CRPR groups was higher (1.86-fold and 1.25-fold, respectively) than that in the CON group, with the number in the CAPR group being the maximum. There were more of the same ASVs between the CAPR and CRPR groups (Figure 2A). As shown in Figure 2B, the number of species in the species accumulation curve reached stability at 25,000 reads, with no further increases, which indicates that the sequencing coverage was saturated. According to the Observed_species value (*p* = 0.034) and Shannon index (*p* = 0.022), the microbial species richness and diversity in the rumen of the dairy cows were significantly higher in the CAPR group than in the CON group, but the differences were not significant between the CRPR and CON groups (Figure 2C). Using Bray–Curtis distance-based PCoA showed that the projection distance of the CAPR and CRPR groups were far from that of the CON group, meaning that Ca-Pro and Cr-Pro had an obvious influence on the composition of the ruminal microbiota of dairy cows (Figure 2D). Adonis analysis further showed the differences between the CAPR, CRPR, and CON groups (*p* < 0.05) (Figure 2E).

### 3.4. The Impact of Cr-Pro and Ca-Pro on Rumen Microbial Composition

A total of 28 phyla, 50 classes, 90 orders, 159 families, 289 genera, and 382 species were identified in the rumen microbiota (Supplementary Data S1). Among the phylum, *Bacteroidetes* (56.64 ± 2.49%), *Firmicutes* (34.21 ± 2.62%), *Proteobacteria* (5.40 ± 2.49%), and *Actinobacteria* (1.69 ± 0.67%) were the dominant bacteria, accounting for more than 90% of the total rumen bacteria (Figure 3A). Sixteen genera (including six unidentified genera) were more than 0.1% in relative abundance, representing the core microbiome among the three groups (Figure 3A and Table 3). *Prevotella* (44.57 ± 2.54%) occupied the highest abundance, but there were no significant intergroup differences (Table 3). In comparison to the CON group, the relative abundance of unidentified*_Bacteroidales* and unidentified*_Ruminococcaceae* showed a significant increase in the CAPR group (*p* = 0.02 and *p* = 0.049, respectively), which were higher in the CRPR group (*p* = 0.053 and *p* = 0.085, respectively). The genera unidentified*_Clostridiales* and *Ruminococcus* were more abundant in the CAPR and CRPR groups, though there were no obvious differences (*p* > 0.05). The abundance of Succiniclasticum and Selenomonas was strongly reduced in the CAPR (*p* = 0.012 and *p* = 0.047, respectively) and CRPR (*p* = 0.105 and *p* = 0.102, respectively) groups. In general, the taxa in CAPR group changed more significantly and were greater in number.

LefSe analysis showed that there were 17 significantly different biomarkers (LDA score > 3) between the CON and CAPR groups. We observed that families *BS11*, *Christensenellaceae*, and *RF16*, genus *RFN20*, and *Clostridium butyricum* sp. were enriched in the CAPR group (*p* < 0.05). Additionally, the phylum *Proteobacteria*, followed by the classes *Gammaproteobacteria* and *Bacilli* and orders *Aeromonadales* and *Lactobacillales* as well as families *Succinivibrionaceae*, *Veillonellaceae*, and *Lactobacillaceae*, genera *Succiniclasticum*, *lactobacillus*, and *Selenomonas*, and *Prevotella copri* sp. were enriched in the CON group (*p* < 0.05) (Figure 4A). Fifteen taxa were identified as distinguishing taxa (LDA score > 2) in the CRPR versus CON group, with the order Alteromonadales, family *Shewanellaceae*, genera *Succinimonas*, *Ruminobacter*, *Caulobacter*, *Bacteroides*, and *Shewanella*, and *Succinimonas amylolytica* sp. being enriched in the CRPR group. Moreover, the cows in the CON group had an enriched relative abundance of the classes *Bacilli* and *Chloroplast*, orders *Lactobacillales* and *Streptophyta*, genus *Lachnospira*, and *Ruminococcus albus* sp. and *Prevotella copri* sp. compared to those in the CRPR group (Figure 4B).

### 3.5. Gene Function Prediction of Rumen Microbial Communities

A total of 169 KEGG gene families were identified using PICRUSt2 software (v2.3.0) to predict gene families based on the 16S rRNA gene sequencing data (Supplementary Data S2). Among these predictions, 14 gene families showed significant differences between the CON and CAPR groups, and 10 gene families between the CON and CRPR groups (Figure 5). Energy metabolism-related functions, such as biosynthesis of unsaturated fatty acids (*p* = 0.023), galactose metabolism (*p* = 0.031), glycolysis/gluconeogenesis (*p* = 0.04), and starch and sucrose metabolism (*p* = 0.044), were significantly increased in the CAPR group (Figure 5A). Compared to the CON group, the carbon fixation pathways in prokaryotes (*p* = 0.022) and starch and sucrose metabolism (*p* = 0.045) were significantly higher in the CRPR group (Figure 5B). Additionally, the gene families involved in antibiotic synthesis functions, specifically streptomycin biosynthesis (*p* = 0.026) and penicillin and cephalosporin biosynthesis (*p* = 0.015), were increased in the CAPR group, while streptomycin biosynthesis (*p* = 0.034) was increased in the CRPR group. It is worth noting that lipopolysaccharide biosynthesis (*p* = 0.047) was reduced in the CAPR group, while flavonoid biosynthesis (*p* = 0.046) was increased in the CRPR group. Biological synthesis-related functions and RNA polymerase were increased in both the CAPR and CRPR groups.

### 3.6. Blood Oxidative Stress Index and Cytokines

The oxidative stress indicators MDA, T-AOC, SOD, and MPO, which are measures of antioxidant index in the blood, were increased in both the CAPR and CRPR groups, with a more pronounced increase observed in the CAPR group (*p* < 0.05) (Table 4). TNF-α, IL-1β, IL-6, and IL-10 were significantly reduced in the CRPR group (*p* < 0.01), while only TNF-α and IL-6 were reduced in the CAPR group (*p* < 0.05) (Table 4).

### 3.7. The Relationship between the Rumen Microbiota, TVFAs, Oxidative Stress Index, Cytokines, and Lactation Performance

Spearman correlation analysis was conducted to examine the relationship between the rumen microbes that changed in the CAPR and CRPR groups, TVFAs, oxidative stress index, cytokines, and lactation performance (Supplementary Data S3). As shown in Figure 6, *g_Lachnospira* exhibited a positive association with four cytokines (r > 0.57, *p* < 0.03) and a negative association with milk yield (r = −0.52, *p* = 0.049). *s_Prevotella_copri* showed a positive association with IL-6 and TNF-α (r > 0.59, *p* < 0.02) and a negative association with milk protein (r = −0.54, *p* = 0.038). *s_Clostridium_butyricum* demonstrated a positive association with MDA, T-AOC, and MPO (r > 0.54, *p* < 0.04). *g_Bacteroides* and *g_Caulobacter* were positively associated with butyrate (r > 0.60, *p* < 0.02) and negatively associated with TNF-α (r < −0.52, *p* < 0.05).

## 4. Discussion

In recent years, the performance and health of dairy cows have been adversely affected by continuous high temperatures in the global environment. As a result, numerous studies have focused on reducing these effects [1,2,3,4,21]. The application of feed additives for the prevention and control of HS is increasing. Several studies have demonstrated that Cr-Pro contributes to reducing the damage and loss of production caused by HS in pigs, broilers, and laying ducks [22,23,24]. Compared to Cr-Pro, Ca-Pro is more widely used in dairy farming but not for HS. There are few reports on the regulation and comparison of the effects of Cr-Pro and Ca-Pro on HS in postpartum dairy cows. Additionally, several recent studies have proposed that HS causes performance degradation by altering the rumen microbiota structure of dairy cows [2,5]. Herein, we hypothesized that Cr-Pro and Ca-Pro could modulate the negative impact of HS on dairy cows by changing the rumen microbiota.

During the study period, the average THI of the barn remained above 68, subjecting the cows to heat stress. Under heat stress conditions, cows reduce body temperature through thermoregulatory mechanisms such as increasing their respiration rate and surface water evaporation, which can also affect performance [1,4]. In this experiment, we found that both Cr-Pro and Ca-Pro significantly reduced the respiratory rate of the HS cows. Moreover, Cr-Pro had a better effect on improving milk yield. Consistently, the concentration of VFAs in the ruminal fluid of cows fed Cr-Pro significantly increased, including acetate, propionate, and butyrate. This is likely because Cr-Pro increased the dry matter intake (DMI) of the dairy cows, resulting in increased VFA production in the rumen and, consequently, higher milk yield. Unfortunately, DMI was not measured in this experiment, but several experiments have shown that supplementing Cr can increase DMI and improve feed efficiency in cows [15,17,25]. Studies by Al-Saiady et al., (2004) and Wu et al., (2021) showed that Cr can increase milk production in dairy cows, which is consistent with the results of this experiment [17,26]. In contrast, Yasui et al., (2013) and Bryan et al., (2004) suggested that the milk yield of dairy cows does not increase after feeding Cr [13,16]. The reason may be that the addition of Cr in these two experiments was relatively low (≤8 mg/day, versus 3.13 g/day in this study), resulting in less obvious effects on milk yield. Feeding Ca-Pro also benefits milk yield and milk quality, such as milk fat, milk protein, and MUN. Furthermore, Ca-Pro achieved a milk fat-to-protein ratio of 1.14, which falls within the optimal range [2]. Martins et al., (2019) also found that Ca-Pro can increase the milk parameters in Holstein cows [27]. Similar to the effect of Cr-Pro, Ca-Pro also increased the concentration of VFAs in the rumen, particularly propionic acid. Based on these findings, we conclude that both Cr-Pro and Ca-Pro can regulate the performance of HS dairy cows by affecting the rumen VFA concentration. However, further research is needed to investigate how Cr-Pro and Ca-Pro affect rumen VFAs.

According to the research of Lin et al., (2019) [28], VFAs and the ruminal microbiome are correlated. In this study, PCoA and Adonis analysis revealed significant differences in the rumen bacterial composition between the experimental groups after sequencing. The composition of the rumen microbiota in the CAPR and CRPR groups was similar, so it is speculated that propionic acid—as a common additive of the two groups—has a greater impact on rumen microbes. Park et al., (2022) and Zhao et al., (2019) both showed that HS can reduce the richness of the rumen bacterial population of dairy cows [3,5]. As a result of this experiment, the alpha diversity indices suggested that the rumen bacterial population was more diverse and richer in HS cows with Ca-Pro. These results indicate that Ca-Pro ameliorates the adverse effects of HS on dairy cows’ ruminal microbiota. Additionally, we confirmed that Ca-Pro has a greater impact on the rumen microbiota compared to Cr-Pro, according to the analysis of microbiota abundance and LEfSe. After feeding Ca-Pro, the number of rumen bacteria and species increased significantly. We speculate that this result may be due to two factors: First, the dosage of Cr-Pro was limited, while a larger amount of Ca-Pro was added. Compared to Cr^3+^ and Ca^2+^, more propionic acid had a greater effect on the structure of the rumen microflora. Second, the ruminal microbiota population may be more sensitive to Ca^2+^, although further research is needed to verify this. Some studies have found that HS increases the relative abundance of phylum *Bacteroidetes* and decreases that of *Firmicutes* [3,5]. In our results, the addition of Ca-Pro and Cr-Pro did not significantly change the abundance of other bacterial taxa.

This study investigated the increase in the abundance of *Ruminococcaceae*, *Clostridiales*, *Ruminococcus,* and *Clostridium butyricum* in the rumen of HS dairy cows after feeding Ca-Pro. These cellulose-degrading bacteria play a critical role in cellulose degradation, absorption, and VFA production in the rumen [2,29,30]. Numerous studies have shown that their numbers decrease in the rumen under HS [2,3,31,32]. This further confirms that HS reduces the feed utilization and performance of dairy cows by inhibiting cellulose-degrading bacteria. The results of feeding Ca-Pro, which increased cellulose-degrading bacteria in the rumen of the HS cows, suggest that Ca-Pro can indirectly improve fiber utilization by affecting the rumen bacteria, thereby increasing the production of VFA in the rumen. Similarly, Cr-Pro also increased the number of cellulose-degrading bacteria, although its effect was weaker. However, Cr-Pro had a more pronounced effect on VFAs, likely by increasing the body’s glucose metabolism and improving DMI, as previously analyzed. Furthermore, after feeding Ca-Pro, the abundance of *amylolytic* and *saccharolytic* bacteria (*Proteobacteria*, *Succinivibrionaceae*, *Succiniclasticum*, and *Selenomonas*) decreased [33]. *Succinivibrionaceae* is the most abundant genus within *Proteobacteria* and, together with *Succiniclasticum* and *Selenomonas*, belongs to propionic acid-generating bacteria that convert succinic acid into propionic acid [29,34]. The increase in the propionic acid concentration in the rumen following Ca-Pro addition may compensate for the inhibition of these bacteria. Uyeno et al., (2010) showed that HS significantly alters feed intake, with cows consuming more concentrate (rather than forage) [35], leading to an increased abundance of soluble carbohydrate-digesting and lactate-producing bacteria [5]. The rise in rumen lactate concentration causes a decrease in rumen pH, reducing cows’ DMI, sorting of preferred feed portions, rumination time, and salivary bicarbonate secretion [2,36]. In this study, the addition of Ca-Pro and Cr-Pro reduced the abundance of *Lactobacillus* (belonging to the *Lactobacillaceae* family) [32], decreasing lactate generation and alleviating HS-induced damage.

A limitation of this study is that the change in rumen lactate concentration was not measured. Based on the decrease in *Lactobacillus* abundance, we speculate that the lactate concentration should decrease accordingly. Further studies are required to confirm this speculation. Adding Cr-Pro increased the abundance of amylolytic bacteria and soluble carbohydrate-utilizing bacteria (*Bacteroides*, *Ruminobacter*, *Succinimonas*, and *Succinimonas amylolytica*) in the rumen [2,13,33,37]. It can be concluded that Cr-Pro not only improves energy metabolism in cows [13] but also promotes carbohydrate utilization by the rumen microbiota, enhancing the energy supply to the body. This is likely the main reason why Cr-Pro significantly enhances milk yield.

In addition, Bernabucci et al., (2002) reported that HS increases the production of oxygen free radicals in dairy cows, and as the oxidative stress level increases, the related antioxidant activities also increase [38]. In this study, the addition of Ca-Pro and Cr-Pro increased blood oxidation and antioxidant indexes (MDA, T-AOC, SOD, and MPO), especially Ca-Pro. Several studies have shown that both propionic acid and chromium have good anti-oxidative stress effects against oxidative stress [24,26,39]. Therefore, it can be concluded that both Ca-Pro and Cr-Pro have obvious anti-oxidative effects. Although Ca-Pro and Cr-Pro can improve the antioxidant capacity of cows, they also enable the body to maintain high production performance under heat stress, which in turn leads to a relatively strong oxidative stress state. This is not contradictory; in the experiment, the added concentration of Ca-Pro was higher than that of Cr-Pro, which may be the reason why the antioxidant index of Ca-Pro increased more noticeably. Oxidative stress causes damage through immune responses such as inflammation. Chen et al., (2018) showed that the levels of cytokines such as TNF-α, IL-6, and IL-1β are higher in the circulation plasma of HS cows [34], further indicating that HS-induced oxidative stress can lead to inflammation [2,40]. In this study, Cr-Pro had a more pronounced alleviating effect on inflammation in the cows as evidenced by the significant decrease in cytokines TNF-α, IL-1β, IL-6, and IL-10 in the blood, which is consistent with the findings of Burton et al., (1996) [41]. Many studies have demonstrated that Cr has a regulatory effect on the immunity of cows, pigs, chickens, and other animals and can inhibit the increase in immune indicators caused by heat stress. However, the specific mechanism is still unclear [17,22,23,41,42]. TNF-α and IL-1β in the blood were also significantly decreased after Ca-Pro supplementation, which is due to the anti-inflammatory advantage of propionic acid, resulting in reduced expression of proinflammatory factors [43]. The results of KEGG gene family predictions showed an increase in the gene families involved in antibiotic synthesis functions after feeding Ca-Pro and Cr-Pro, which partly explains the decrease in inflammatory cytokines in the blood of the cows. It is known that lipopolysaccharide (LPS) from Gram-negative bacteria in the rumen is the main cause of inflammation [44,45]. However, lipopolysaccharide biosynthesis was inhibited in the Ca-Pro feeding group, and flavonoid biosynthesis, which has antioxidant effect, was enhanced in the Cr-Pro group [46]. We should note that we did not focus on the changes of these substances in the rumen, and these findings need further confirmation. Our experiment demonstrated that Ca-Pro and Cr-Pro have good anti-oxidative stress and anti-inflammatory effects on heat-stressed cows, but there may be other effects that should also be the focus of further study.

Gene function prediction also revealed that energy metabolic pathways and biological synthesis-related functions were increased in both the Ca-Pro and Cr-Pro groups. Chen et al., (2018) found that HS results in a downregulation of the metabolic pathways in dairy cows [34], whereas our results suggest that Ca-Pro and Cr-Pro can mitigate the negative effects of HS in this regard. Although the specific regulation mechanisms of these effects of the Ca-Pro and Cr-Pro are not yet fully understood, they provide important reference value for future research directions. Spearman correlation analysis indicated that *Lachnospira* and *Prevotella_copri* in the rumen are positively associated with cytokines. This is similar to the finding of Chen et al., (2021) that *Prevotella_copri* activates the host chronic inflammatory response through the TLR4 and mTOR signaling pathways [47]. However, Wanget al., (2021) suggests that *Lachnospira* can suppress inflammation in early-lactation dairy cows [30]. Additionally, this study found that *Bacteroides* and *Caulobacter* are positively associated with butyrate and negatively associated with TNF-α. We conclude that they may reduce TNF-α by increasing the concentration of butyric acid, taking advantage of its anti-inflammatory effect. The rumen microflora is affected by various factors and interacts with the host to produces a variety of physiological effects, making it a complicated process. To determine the specific mechanism of action of Ca-Pro and Cr-Pro on rumen bacteria, we will utilize more advanced sequencing methods and conduct related experiments in future studies.

## 5. Conclusions

In summary, we conclude that the application of Cr-Pro or Ca-Pro can provide relief for HS dairy cows, but their modes of action and outcomes differ. Ca-Pro exhibits a stronger regulatory effect on the rumen microbiota and dairy quality, while Cr-Pro primarily focuses on regulating the body’s energy metabolism, leading to increased milk yield and anti-inflammatory effects. It is worth emphasizing that both have their advantages, but the use of Cr-Pro in production is limited. Therefore, we recommend using both to alleviate HS in dairy cows, which will be the focus of our research in the next phase.

## Figures and Tables

**Figure 1 microorganisms-11-01625-f001:**
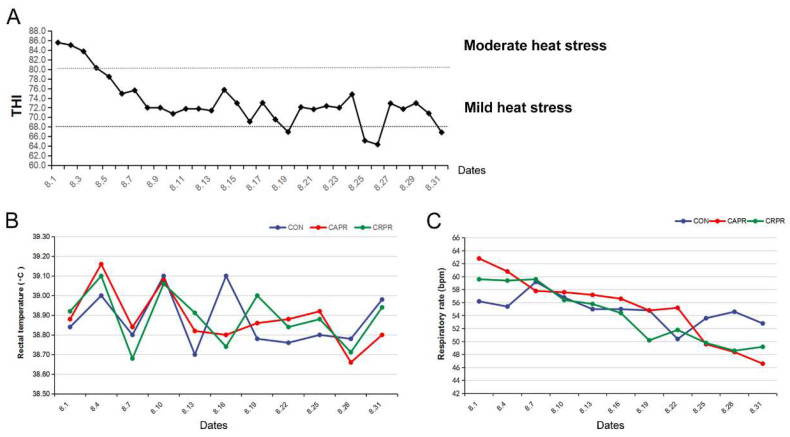
(**A**) The temperature–humidity index (THI). The value in the dashed line range was 68 to 80, represented mild heat stress of cows. (**B**) Shows rectal temperature (°C) of the three groups. (**C**) The respiratory rate (beats per minute, bpm) of the three groups. Dates were from 1 August 2020 to 31 August 2020. CON, control group; CAPR, calcium propionate group; CRPR, chromium propionate group. *n* = 5 in each group.

**Figure 2 microorganisms-11-01625-f002:**
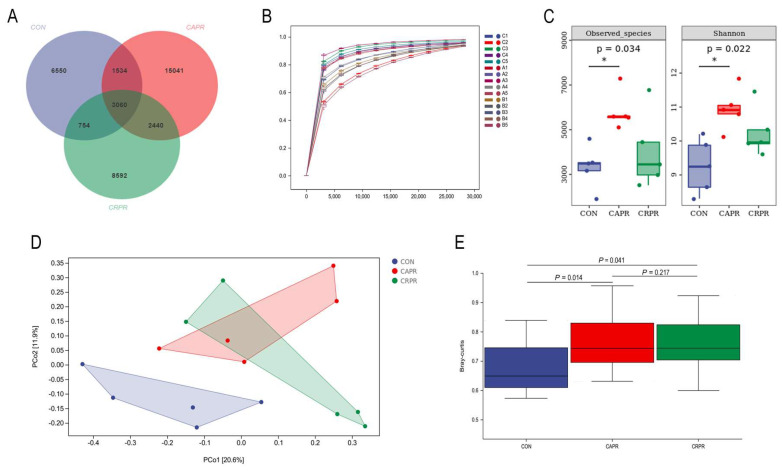
(**A**) ASVs-Venn diagram analysis of CON, CAPR, and CRPR. (**B**) Species accumulation curve analysis (CON includes C1–C5 samples, CAPR includes A1–A5 samples, and CRPR includes B1–B5 samples). (**C**) Observed_species and Shannon indices. (**D**) The principal coordinates analysis (PCoA) profiles of ruminal microbiota based on Bray–Curtis. (**E**) Adonis analysis. CON, control group; CAPR, calcium propionate group; CRPR, chromium propionate group. * was *p* < 0.05. *n* = 5 in each group.

**Figure 3 microorganisms-11-01625-f003:**
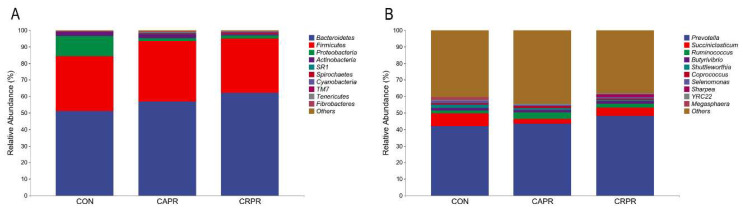
The rumen microbiome composition across the CON, CAPR, and CRPR. (**A**) The top 10 microbiome at phylum level. (**B**) The genus levels. CON, control group; CAPR, calcium propionate group; CRPR, chromium propionate group. *n* = 5 in each group.

**Figure 4 microorganisms-11-01625-f004:**
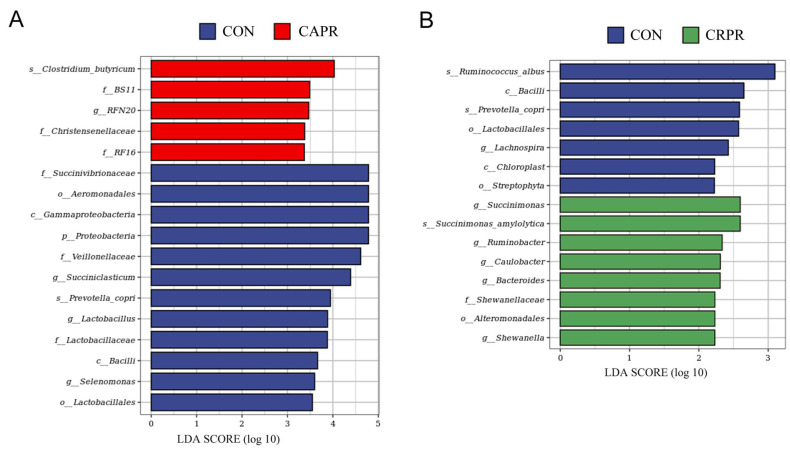
The LDA effect size (LEfSe) analysis of bacterial taxa. (**A**) CON versus CAPR. (**B**) CON versus CRPR. Significant differences are defined as LDA value > 2 and *p* < 0.05. The length of the bar chart represents the influence on different species. CON, control group; CAPR, calcium propionate group; CRPR, chromium propionate group. *n* = 5 in each group.

**Figure 5 microorganisms-11-01625-f005:**
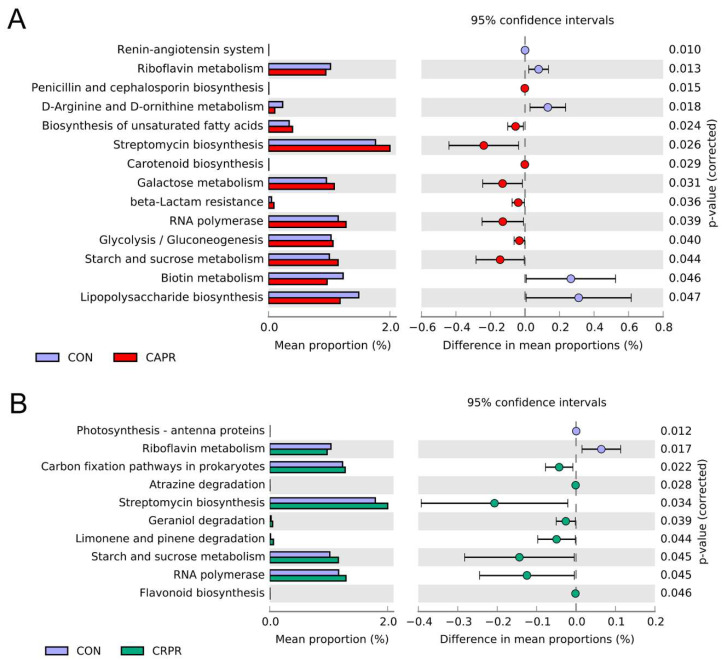
PICRUSt2 functional predictions based on the Kyoto Encyclopedia of Genes and Genomes (KEGG). (**A**) CON versus CAPR. (**B**) CON versus CRPR. The bars on the left represent the abundance ratio of each function in both sets of groups, the chart on the right shows the percentage of difference in functional abundance, and P-values are displayed on the far right of each panel. CON, control group; CAPR, calcium propionate group; CRPR, chromium propionate group. *n* = 5 in each group.

**Figure 6 microorganisms-11-01625-f006:**
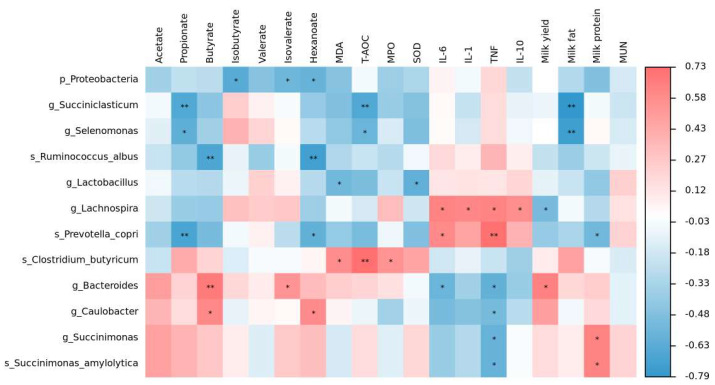
Spearman correlation analysis between rumen microbiota, TVFAs, oxidative stress index, cytokine, and lactation performance. The rows represent the microbe, the columns represent the other index, and each colored box represents a Spearman correlation coefficient. Red represents the positive correlation, and blue represents the negative correlation. * *p* < 0.05, ** *p* < 0.01.

**Table 1 microorganisms-11-01625-t001:** The basal total mixed ration (TMR) formulation.

Feed Composition	(% of TMR)	Nutrient Composition	(% of DM) ^2^
Leymus chinensis	2.80	CP	16.48
RUP	1.17	ME (mCal/kg)	2.68
Corn silage	63.08	NEL (mCal/kg)	1.72
Premix ^1^	5.84	ADF	17.31
Corn	12.85	NDF	29.52
Soybean meal	6.78	Crude fat	4.75
Corn germ meal	3.97	Ca	0.79
Cottonseed	3.51	K	1.09
Total	100.00	Na	0.42

^1^ The premix provided the following per kg of diets: Fe 2000 mg, Zn 2400 mg, Cu 5000 mg, Mn 2180 mg, Se 150 mg, Co 40 mg, I 250 mg, Vitamin A 2000 kIU, Vitamin D 500 kIU, and Vitamin E 5700 kIU. ^2^ Percent of diet dry matter. Note. RUP: ruminally undegraded protein; CP: crude protein; ME: metabolizable energy; NEL: net energy for lactation; ADF: acid detergent fibre; NDF: neutral detergent fiber.

**Table 2 microorganisms-11-01625-t002:** The milk yield, milk composition, and ruminal parameters in the CON, CAPR, and CRPR groups of cows.

	Treatment ^1^				
Items	CON	CAPR	CRPR	SEM	*p*-Value
Milk yield (kg/d)	38.09 ^b^	40.60 ^b^	51.35 ^a^	2.17	0.018
Milk fat (%)	3.13	3.75	3.43	0.20	0.487
Milk protein (%)	3.14	3.31	3.26	0.07	0.585
Milk fat:protein	0.99	1.14	1.06	0.06	0.626
MUN ^2^ (mg/dL)	16.52	17.92	18.82	0.80	0.521
Ruminal TVFAs ^3^, mM					
Total	46.61 ^b^	50.98 ^ab^	61.78 ^a^	2.60	0.034
Acetate	23.71 ^b^	22.70 ^b^	30.21 ^a^	1.50	0.073
Propionate	11.99 ^b^	16.63 ^ab^	17.51 ^a^	1.08	0.067
Butyrate	7.19 ^b^	7.73 ^ab^	9.85 ^a^	0.51	0.066
Isobutyrate	0.69	0.69	0.70	0.04	0.993
Valerate	1.22	1.06	1.19	0.06	0.548
Isovalerate	1.46	1.55	1.74	0.09	0.496
Hexanoate	0.34	0.63	0.57	0.08	0.289
A:P Ratio ^4^	2.02	1.41	1.79	0.13	0.173

^1^ CON, control group; CAPR, calcium propionate group; CRPR, chromium propionate group. *n* = 5 in each group. ^2^ MUN, milk urea nitrogen; ^3^ TVFAs, total volatile fatty acids; ^4^ The average ratio of acetate to propionate. ^a,b^ Means with different superscript letter differ (*p* < 0.05) within groups.

**Table 3 microorganisms-11-01625-t003:** Phylum and genus composition of the rumen bacteria in the CON, CAPR, and CRPR groups cows.

		Abundance ^1^ (%)				
Phylum	Genus	CON	CAPR	CRPR	SEM	*p*-Value
*Bacteroidetes*		51.14	56.73	62.04	2.49	0.212
	*Prevotella*	42.05	43.37	48.28	2.54	0.608
	unidentified_*Bacteroidales*	1.98 ^b^	5.21 ^a^	4.60 ^ab^	0.59	0.049
	*YRC22*	0.58	0.82	0.62	0.10	0.642
*Firmicutes*		33.12	36.69	32.80	2.62	0.821
	unidentified *Ruminococcaceae*	3.60 ^b^	7.89 ^a^	7.26 ^ab^	0.90	0.100
	unidentified_*Clostridiales*	3.87	9.27	4.74	1.31	0.275
	*Succiniclasticum*	7.62 ^a^	2.99 ^b^	4.86 ^ab^	0.78	0.037
	*Ruminococcus*	1.81	3.99	2.34	0.61	0.334
	*Butyrivibrio*	1.40	1.34	1.58	0.16	0.841
	*Shuttleworthia*	2.18	1.15	0.83	0.34	0.357
	unidentified *Lachnospiraceae*	1.20	1.49	1.24	0.12	0.571
	*Coprococcus*	0.98	1.23	0.79	0.14	0.651
	*Selenomonas*	1.23 ^a^	0.53 ^b^	0.67 ^ab^	0.15	0.105
	*Sharpea*	0.13	0.14	1.82	0.59	0.883
	*Megasphaera*	1.58	0.03	0.02	0.51	0.320
*Proteobacteria*		12.33	1.93	1.94	2.49	0.102
	unidentified *Succinivibrionaceae*	11.97	1.63	1.39	2.50	0.065
*Actinobacteria*		2.15	2.35	0.56	0.67	0.519
	unidentified *Bifidobacteriaceae*	1.79	2.11	0.28	0.64	0.496

^1^ Main bacteria relative abundances >0.1% among three groups. CON, control group; CAPR, calcium propionate group; CRPR, chromium propionate group. *n* = 5 in each group. ^a,b^ Means with different superscript letter differ (*p* < 0.05) within groups.

**Table 4 microorganisms-11-01625-t004:** The concentration of blood oxidation, antioxidation, and cytokine in the CON, CAPR, and CRPR groups of cows.

	Treatment ^1^				
Items	CON	CAPR	CRPR	SEM	*p*-Value
MDA ^2^ (nM/mL)	4.52 ^b^	6.13 ^a^	5.23 ^ab^	0.27	0.032
T-AOC ^3^ (U/mL)	13.76 ^b^	17.32 ^a^	16.08 ^ab^	0.59	0.030
SOD ^4^ (U/mL)	22.93 ^b^	24.48 ^a^	23.72 ^ab^	0.32	0.131
MPO ^5^ (U/L)	199.82 ^b^	238.41 ^a^	216.28 ^ab^	7.22	0.081
Blood Cytokine (pg/mL)					
TNF-α ^6^	271.94 ^a^	244.68 ^b^	142.13 ^c^	11.37	0.000
IL-1β ^7^	674.42 ^a^	619.93 ^ab^	576.21 ^b^	13.28	0.006
IL-6 ^8^	251.57 ^a^	229.67 ^b^	209.96 ^c^	4.74	0.000
IL-10 ^9^	104.45 ^a^	96.69 ^ab^	92.26 ^b^	2.16	0.061

^1^ CON, control group; CAPR, calcium propionate group; CRPR, chromium propionate group. *n* = 5 in each group; ^2^ MDA, malonaldehyde; ^3^ T-AOC, total antioxidant capacity; ^4^ SOD, superoxide dismutase; ^5^ MPO, myeloperoxidase; ^6^ TNF-α, tumor necrosis factor-α; ^7^ IL-1β, interleukin-1β; ^8^ IL-6, interleukin-6; ^9^ IL-10, interleukin-10. ^a,b,c^ Means with different superscript letter differ (*p* < 0.05) within groups.

## Data Availability

Data from the sequencing were deposited in the Sequence Read Archive (SRA) of NCBI (No. SRR416794: PRJNA922454).

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
