# Peer review of "Effects of Chromium Propionate and Calcium Propionate on Lactation Performance and Rumen Microbiota in Postpartum Heat-Stressed Holstein Dairy Cows"

_microorganisms, 2023, doi:10.3390/microorganisms11071625_

Round 1

Reviewer 1 Report

The manuscript entitled “Effects of chromium propionate and calcium propionate on lactation performance and rumen microbiota of postpartum heat stressed Holstein dairy cows” brings new and very interesting data concerning changes in lactation parameters, rumen microbiota and immune system of dairy cows according to different food supplements with the emphasis on their influence during heat stress. The study design is rationale, the methods are properly chosen, the results are clearly presented and the conclusions are justified by them. My questions and remarks concerning the manuscript are listed below:

-          Line 37 – describe HS when using for the first time   

-          Lines 52, 57 – please describe the abbreviations

-          Line 61 – explain ‘perinatal lactation’

-          Lines 84-87 – how were Ca-Pro and Cr-Pro given to the cows, was it mixed in TMR or other way?

-          Line 170 – parameters were

-          Table 2 – were the lactation parameters the average of three milkings at the last day or the average of the whole study period? Did you measure milk yield at the beginning of study period because it could also influence the production?

-          Line 254 – delete ‘There were’ and start with ‘Sixteen…’

-          Line 271 – ‘Fifteen…’

-          Line 362 – English

Taking into consideration all the above in my opinion the manuscript is suitable to be published in Microorganisms after minor corrections and expalanations.

Author Response

We would like to thank the reviewers for providing us constructive suggestions that have helped us improve the quality of the paper. Herein, we submit a new version of our manuscript with the title “Effects of chromium propionate and calcium propionate on lactation performance and rumen microbiota of postpartum heat stressed Holstein dairy cows” (ID: 2446238). The manuscript has been modified according to the reviewers’ suggestions. The following is a point-by-point response to the reviewers’ comments.

The manuscript entitled “Effects of chromium propionate and calcium propionate on lactation performance and rumen microbiota of postpartum heat stressed Holstein dairy cows” brings new and very interesting data concerning changes in lactation parameters, rumen microbiota and immune system of dairy cows according to different food supplements with the emphasis on their influence during heat stress. The study design is rationale, the methods are properly chosen, the results are clearly presented and the conclusions are justified by them. My questions and remarks concerning the manuscript are listed below:

- Line 37 – describe HS when using for the first time

Response: Thanks for the reviewer’s suggestion. We have inserted it on Line 33.

- Lines 52, 57 – please describe the abbreviations

Response: We have revised them on Line 47 and Line 52.

- Line 61 – explain ‘perinatal lactation’

Response: We have revised this sentence on Line 55-57: Studies have shown that chromium supplementation improves milk yield in early lactation (21 days after delivery), and modulates immunity and reproduction in dairy cows.

- Lines 84-87 – how were Ca-Pro and Cr-Pro given to the cows, was it mixed in TMR or other way?

Response: Ca-Pro and Cr-Pro were mixed in TMR to give the cows. We have revised this description on Line 86-90.

- Line 170 – parameters were

Response: We have added them on Line 168: The plasma parameters (oxidation, antioxidation and cytokine), lactation performance (milk yield, fat, protein and MUN) and rumen fermentation parameters (VFAs)...

- Table 2 – were the lactation parameters the average of three milkings at the last day or the average of the whole study period? Did you measure milk yield at the beginning of study period because it could also influence the production?

Response: “Milk production was recorded by Afimilk MPC system (Afimilk Agricultural Science and Technology Co., Ltd., Israel) ,and the recorded value was the average of three times on the same day. The average milk yield of the first three days of the experimental period was calculated as the initial milk yield of the cows, and the average milk yield of the last three days for comparison of results”. We added the above content on Line 127-132, and added the initial milk yield on Line 86-90

- Line 254 – delete ‘There were’ and start with ‘Sixteen…’

Response: Thanks for the suggestion. We have revised it on Line 252.

- Line 271 – ‘Fifteen…’

Response: Thanks for the suggestion. We have revised it on Line 270.

- Line 362 – English

Response: Thanks for the suggestion. We have revised this part of the discussion.

Taking into consideration all the above in my opinion the manuscript is suitable to be published in Microorganisms after minor corrections and expalanations.

Reviewer 2 Report

Effects of chromium propionate and calcium propionate on lactation performance and rumen microbiota of postpartum heat stressed Holstein dairy cows

First, please proof-read your manuscript for wording, grammar, and syntax errors using an English Service center or native scientist in the area. I pointed out few corrections but it is not limited to my notes, as there are many more errors in wordings, phrases and grammar. Please take action to resolve this issue.

The manuscript is original and worth investigation. The Abstract and Introduction need some improvements as I stated below. Discussion should be re-written in whole and some obtained results missed to discussed.

Abstract

L19: Please specify the cow conditions (e.g. BW, Milk yield, DIM)

L24: Please remove: “The results shoed that”

Introduction

L35: Please remove the first sentence as it does not bring any related knowledge.

L51: Remove “widely”

L52-54: Please re-write the sentence

L54-55: The sentence is unclear. Please re-write it

L57: Do you mean another propionate source? Additive? Chelate? Please clarify

L60-61: The sentence is not grammatically correct

L65: A growing number of investigations have shown the beneficial effects of Ca-Pro and Cr-Pro in production, which mainly focus on energy metabolism, growth and reproductive performance

Where are the references for this statement?

M&M

The milk yield average is not indicated for each group

L78: Formatting of the subtitle (capital wording) need to be consistent)

L83: Please italicize the “ad libitum”

Table 1: Please re-write the title of the table. also (Dry matter  basis should be removed as it is stated in the table row

L97: The verb is missing so the sentence is broken

L115: from the tail using venipuncture….Please re-write

L125-126: Need revision sentence

Results

Table 2: All values should have two decimal points to be consistent (except p values which is correct in the way presented).

Discussion

Please check the English flow, grammar, wording for all sentences used.

P values are already stated in the Results section, so it should not be used in Discussion

The first paragraph of discussion need to be reduced to few sentences as it is reporting the necessity of the experiment which is already stated in Introduction

L392: Citation formatting issue

Discussion should be reorganized and expanded. Some of the results are not discussed at all. Please re-write the section using the existing input plus adding the discussion and speculation of the all results obtained.

The English need extensive modifications in whole. Lots of wording, grammatical and syntax errors can be seen thoroughly.

Author Response

We would like to thank the reviewers for providing us constructive suggestions that have helped us improve the quality of the paper. Herein, we submit a new version of our manuscript with the title “Effects of chromium propionate and calcium propionate on lactation performance and rumen microbiota of postpartum heat stressed Holstein dairy cows” (ID: 2446238). The manuscript has been modified according to the reviewers’ suggestions. The following is a point-by-point response to the reviewers’ comments.

Effects of chromium propionate and calcium propionate on lactation performance and rumen microbiota of postpartum heat stressed Holstein dairy cows

First, please proof-read your manuscript for wording, grammar, and syntax errors using an English Service center or native scientist in the area. I pointed out few corrections but it is not limited to my notes, as there are many more errors in wordings, phrases and grammar. Please take action to resolve this issue.

The manuscript is original and worth investigation. The Abstract and Introduction need some improvements as I stated below. Discussion should be re-written in whole and some obtained results missed to discussed.

Abstract

L19: Please specify the cow conditions (e.g. BW, Milk yield, DIM)

Response: Thanks for the reviewer’s suggestion. We have added the cow conditions on L17.

L24: Please remove: “The results shoed that”

Response: We have removed the words on L22.

Introduction

L35: Please remove the first sentence as it does not bring any related knowledge.

Response: We have removed this sentence.

L51: Remove “widely”

Response: We have removed this word.

L52-54: Please re-write the sentence

Response: We have revised this sentence on L46-48: Notably, calcium propionate (Ca-Pro) is widely used as a preservative in food, feed, and medicine due to its efficacy against mycetes and bacteria, and its neutral taste and flavor.

L54-55: The sentence is unclear. Please re-write it

Response: We have revised this sentence on L49-51: In the rumen, Ca-Pro is hydrolyzed to propionate and Ca2+, which are both important components of rumen fluid.

L57: Do you mean another propionate source? Additive? Chelate? Please clarify

Response: We modified these sentences on L52-54 to make them better understood: “Furthermore, chromium propionate (Cr-Pro), another form of propionate, has been gradually used as a feed additive. It plays a crucial role in the metabolism of carbohydrates, lipids, and proteins”.

L60-61: The sentence is not grammatically correct

Response: We have revised this sentence on L55-56: Studies have shown that chromium supplementation improves milk yield in early lactation (21 days after delivery), and modulates immunity and reproduction in dairy cows.

L65: A growing number of investigations have shown the beneficial effects of Ca-Pro and Cr-Pro in production, which mainly focus on energy metabolism, growth and reproductive performance

Where are the references for this statement?

Response: We have added the references on L67.

M&M

The milk yield average is not indicated for each group

Response: We have added them on L85-L89: Control group (CON, n = 5; milk yield, 41.2 ± 0.85 kg/day), fed the basal total mixed ration (TMR) diet (Table 1); Ca-Pro supplementation group (CAPR, n = 5; 40.7 ± 1.38 kg/day), fed the basal TMR mixed with 200 g/day of Ca-Pro; Cr-Pro supplementation group (CRPR, n = 5, 41.8 ± 1.21 kg/day), fed the basal TMR mixed with 10 g/day of Cr-Pro.

L78: Formatting of the subtitle (capital wording) need to be consistent

Response: We have revised it on L78: 2.2 Animals, Diets, and Experimental Design.

L83: Please italicize the “ad libitum”

Response: We have revised it on L83.

Table 1: Please re-write the title of the table. also (Dry matter basis should be removed as it is stated in the table row

Response: Thanks for your suggestion. We have revised the table 1 and it’s title on L99.

L97: The verb is missing so the sentence is broken

Response: We have revised this sentence on L97: “The formula for calculating the temperature and humidity index was THI = 0.81 T + (0.99 T – 14.3)RH + 46.3”.

L115: from the tail using venipuncture….Please re-write

Response: We have revised this sentence on L112.

L125-126: Need revision sentence

Response: We have revised this sentence on L122-123: Milk samples were collected from all 15 cows three times on the 31st day. For each collection, 50 mL of milk was taken after thorough mixing.

Results

Table 2: All values should have two decimal points to be consistent (except p values which is correct in the way presented).

Response: Thanks for your suggestion. We have revised the table 2.

Discussion

Please check the English flow, grammar, wording for all sentences used.

Response: Thanks to the reviewer's suggestion, we have revised the discussion as a whole, including structure, sentence, grammar and wording.

P values are already stated in the Results section, so it should not be used in Discussion

Response: We have deleted the p values in discussion.

The first paragraph of discussion need to be reduced to few sentences as it is reporting the necessity of the experiment which is already stated in Introduction

Response: We have thoroughly revised The first paragraph of discussion and integrated part of it into the introduction. (L346-356)

L392: Citation formatting issue

Response: We have revised it on L381: According to the research of Lin et al. (2019) [28], VFAs and the ruminal microbiome are correlated.

Discussion should be reorganized and expanded. Some of the results are not discussed at all. Please re-write the section using the existing input plus adding the discussion and speculation of the all results obtained.

Response: We have reorganized and expanded the discussion. And we have marked the modified and expanded parts in red for reviewers to review.

The English need extensive modifications in whole. Lots of wording, grammatical and syntax errors can be seen thoroughly.

Response: The manuscript had undergone English language editing by MDPI (ID: english-67398). The text has been checked for correct use of grammar and common

technical terms, and edited to a level suitable for reporting research in a scholarly journal. MDPI uses experienced, native English speaking editors. Full details of the editing service can be found at https://www.mdpi.com/authors/english.

Round 2

Reviewer 2 Report

No additional comments are required. 

Minor proof-read needed